# An Impact Assessment of GHG Taxation on Emilia-Romagna Dairy Farms through an Agent-Based Model Based on PMP

**Lisa Baldi** [1] , **Filippo Arfini** [2,*] , **Sara Calzolai** [2] **and Michele Donati** [1]

[1]  Department of Chemistry, Life Sciences and Environmental Sustainability, University of Parma, 43124 Parma, Italy; lisa.baldi@unipr.it (L.B.); michele.donati@unipr.it (M.D.)
[2]  Department Economic and Management, University of Parma, 43125 Parma, Italy; sara.calzolai@unipr.it
*  Correspondence: filippo.arfini@unipr.it

**Abstract:** The aim of this work is to assess the structural, production, environmental, and economic impact of an increasing tax on climate change gas emissions related to milk production under the current CAP payment system. The analysis is performed using an Agent-Based Model (ABM) based on Positive Mathematical Programming (PMP). The integration between ABM and PMP makes it possible to simulate farmers' strategies considering the interaction between them, the territorial specificity, and the heterogeneity of farms in the presence of little information on production costs. It also makes it possible to add a social and cultural perspective to the economic factors. The model is calibrated using FADN data for the Emilia-Romagna region (Italy) from the year 2020. The results show that farmers belonging to different age groups make decisions based on economic profitability, but also on their social and cultural background. To maximise their utility functions, farmers can opt for more efficient agricultural management practices that may result in the exchange of production factors, especially land. The overall impact penalises less efficient farms and agricultural production with higher negative externalities.

**Keywords:** CAP reform; $CO_2$ taxation; agent-based model; PMP

## 1. Introduction [1]

The demand for food is expected to increase in the next few years and the need for a more sustainable livestock sector can no longer be ignored. Worldwide, livestock is responsible for 16.5% of all anthropogenic GHG emissions, mainly in the form of methane ($CH_4$), carbon dioxide ($CO_2$), nitrous oxide ($N_2O$), and ammonia ($NH_3$), coming from fodder cultivation, enteric fermentation, manure management, and nitrogen deposition and application [2,3]. Livestock activities require an extensive amount of land, both for accommodating animals and for fodder cultivation. This generally translates to deforestation, leading to emissions of the carbon previously stored in biomass and in soil. Furthermore, the livestock sector is a great cause of waste and the pollution of water, which is contaminated by animal excreta, antibiotics and hormones, fertilizers and pesticides used in forage production, and runoff from pasture [4]. Water quality degradation, eutrophication, and hypoxia in surface water bodies are mainly due to the nitrogen and phosphorous input coming from livestock manure management and fertilizers [5].

In 2020, the Italian livestock sector produced 271,051 thousand tonnes of ammonia and 19,760 thousand tonnes of $CO_2eq$, 68% of which (13,535 thousand tonnes) was related to cattle enteric fermentation and cattle manure management. The national animal production value (in current value) accounted for 15.5 billion EUR [6]. The Emilia-Romagna region is responsible for 10.4% of Italian livestock-related GHG emissions (2059 thousand tonnes) [7], and its economy heavily relies on the Parmigiano Reggiano industry.

Since the MacSharry Reform in 1992, the European Common Agricultural Policy has evolved to ensure food security through more sustainable agricultural practices [8]. Eco-schemes (ES) and Agriculture Environmental Schemes (AES) have been introduced as new

policy tools with the post 2020 CAP reform to align the CAP objectives to the European Green Deal targets: reaching climate neutrality by 2050 and halving fertilizer application and nutrient loss by 2030 [9]. Despite the introduction of these new mechanisms, the latest strategies proposed by the European Commission in response to the recent global food insecurity issues caused by the war in Ukraine are likely to have a negative environmental effect. Suffice it to mention the exemption from the greening obligations and permission to cultivate on fallow land that falls within the Ecological Focus Areas (EFAs) [10].

In recent decades, the European Union has also developed a carbon pricing system to reduce GHG emissions and mitigate climate change. The two main carbon pricing mechanisms implemented so far are the Emissions Trading Systems (ETS) and carbon taxes. The ETS was set up in 2005 (Directive 2003/87/EC) as a cap-and-trade approach for activities, which are required to have allowances equivalent to their emissions. However, agricultural activities are not yet included in these carbon pricing mechanisms [11]. Policy makers have been reluctant to include them, partly because of a lack of political will, and partly because of the difficulty of measuring emissions and emission reductions at the farm level [12].

Carbon taxes, on the other hand, directly set a price on carbon emissions, with the aim of incentivising activities to reduce their emissions. Finland was the first EU country to apply a carbon tax in 1990. Carbon taxes are not compulsory for Member States, and the amount applied can vary widely: from more than 100 EUR/t$CO_2$eq in the northern countries, to less than 1 EUR in Poland and Ukraine [13].

In Italy, a carbon tax was introduced in 1999 (L 448/1998) on the consumption in energy plants of coal, petroleum, and coke, with a tax rate initially fixed at 1000 £/t of product (around 0.52 EUR/t), but it was in force only for that year. Since 1999, the reintroduction of the carbon tax in Italy has been discussed but not reimplemented [14].

In 14 July 2021, the Commission published the "Ready for 55%" package, setting out the Green New Deal [15]. It includes the revision of the ETS Directive and introduces the Carbon Border Adjustment Mechanism (CBAM) to prevent carbon leakage and encourage a global move towards net zero carbon emissions, in line with the Paris Agreement. The CBAM regulation was approved by the Council in March 2022 [16].

Applying measures to decrease the GHG emissions of the agricultural sector could significantly reduce the ecological footprint of agriculture, but could also negatively affect farm competitiveness and incomes [17]. This negative effect could, however, be mitigated if farmers, operating in the same context, could engage in the exchange of production factors to reduce the inefficiencies of single farms.

The aim of this research is to assess, ex-ante, how farmers could react to the opportunity to apply to eco-schemes to make up for potential revenue losses, in an environment where they can reduce their inefficiency by exchanging production factors such as land and pollution quotas. The effect of an increasing carbon taxes (20, 50, 100, and 150 EUR/t$CO_2$eq) is evaluated to simulate farmers' responses in terms of changing production plans and resource allocation.

To reproduce a complex environment at the farm scale, in which farmers can interact with each other while maximizing their farms' utility function, a Dairy Farm Agent-Based Model (AGRISP), based on Positive Mathematical Programming (PMP), is developed and applied. ABMs make it possible to evaluate agricultural policies and farmers' level of acceptance by simulating interactions between farmers and, at the same time, taking subregional territorial specificity and farm heterogeneity into account. PMP methodology makes it possible to add social and cultural perspectives to economic drivers [18,19].

The integration of an ABM and the PMP approach into AGRISP (Agricultural Regional Integrated Simulation Package) make it possible to optimize every farm cost function in the sample, taking into account individual farmers' behaviour and characteristics, starting from the observed optimal situation to simulate structural changes, such as changes in farm dimensions or a possible abandonment of farm activity. The model can estimate this choice by simulating exchanges of resources, as well as the introduction of new activities and

changes into agricultural management practices. The aggregation of the regional results can also provide useful and solid insight into the general trend of the agricultural sector at the national and international level.

The paper is organized as follows. Section 2 presents the characteristics of the Agent-Based model developed through a Positive Mathematical Programming approach, the sample data used for the simulation, and the policy scenarios. Section 3 presents the obtained results, while Sections 4 and 5 conclude and suggest paths for future research.

## 2. Materials and Methods

### 2.1. ABM and PMP

Since 2013 and the introduction of the CAP greening measures, researchers have increasingly focused on farm models based on microeconomic data. These models are designed to show individual farmers' behaviour in reacting to market evolution scenarios, agricultural policies, changes in technology sets, and climate change. Recent studies [18,20,21] have found that ABMs are better suited to assess policies holistically, considering environmental, social, and agricultural aspects and, at the same time, making innovations to the mathematical programming used for evaluating agricultural policies. ABMs are models composed by a set of decision makers (the agents) and an environment in which these agents interact with each other. They require rules to define the relationships between the agents and the relationships between the agents and their economic and bio-physical environments, as well as rules defining the sequence of actions occurring in the model [22]. Agent-based models are applied widely in many fields, such as consumer behaviour [23], travel forecasting [24], and disease-spreading control [25]. In agro-economics, they have been heavily used for simulating land-use choices based on an agent's utility derived from land [26,27] and as tool to explore the potential of landscapes to provide multiple ecosystem services [28]. The acting agent, with pre-defined behavioural rules set at the individual farmers' level, seems to be the appropriate starting point for explaining or predicting the choices between different options [29]. According to Möhring, "the great achievement of agent-based models is their integration of the heterogeneity of individuals and transactions, accomplished by placing the optimisation process back on the unit where it actually occurs" ([29] page 10). Like mathematical programming (MP) farm models, ABMs can represent agents' behaviour regarding their production choices: what products to market, what technologies to adopt, what production factors to use (land, labour, and water, etc.), and in what quantities. In AGRISP, the agents are represented by individual farm holders who exchange production factors, mainly land, but also rights to pollute (nitrate spreading) [30]. The model optimizes, simultaneously, the production factors' allocation of each agent, thus the decisions of each impact the behaviour and decisions of the others, simulating how their factor endowment evolves due to production factor exchange or policy impacts. AGRISP captures farms' heterogeneity in terms of farm structure and production strategies, but also in terms of the interactions between farms in the use of scarce resources and evaluating structural changes under the assumption of not-fully rational production choices, maximizing the utility function rather than the profit function [18,31]. Agents, and the environment in which they operate, are defined based on their characteristics. The agents' attributes considered are the age of the farm holder and the presence of heirs. As far as the environment is concerned, altitude and agrarian regions are considered. Individual attributes trigger behavioural rules, more precisely:

- farm holders older than 65 and with no successor receive a monthly retirement pension and do not rent additional land;
- farmers located in NVZ limit manure spreading to 170 kg per hectare, whereas elsewhere, this limit is set to 340 kg/ha, and in both cases, farmers are pushed to rent out their land if unused.

AGRISP represents farmers not just as individual entrepreneurs, but rather as farm-householders, with room in the decision-making process for mediation between family members, which may generate economic inefficiencies [32,33]. Agents decide on the basis of endowment factors, technological knowledge, and individual perception of the economic

and technical risks, and these decisions represent the agents' optimal economic choice. This representation is possible because agent-based farm models can, in sum:

- consider the individual farms and farm-households' heterogeneity,
- reproduce production choices based on the observed activities,
- depict the production specializations and the technologies used.

The literature counts some attempts to assess the effect of the CAP measures through ABMs, such as AgriPoliS [34], MP-MAS [35], LUDAS [36], RegMAS [37], and SWISSland [29].

Normative MP models, such as AROPAj [38], are well suited to assess what production system changes are needed to reduce GHG emissions [39], but are not appropriate to represent agents as described above, as they assume that farmer behaviour is fully rational and do not correctly estimate all of a farmer's explicit and implicit costs. Empirical evidence has shown that solutions obtained using the normative MP model calibration phase differ from the observed data [40–42].

Positive mathematical programming (PMP) models, on the other hand, are based on the assumption that the observed production level, reproduced in the calibration phase, is the result of the optimal agent choices. Some examples of well-established models, based on PMP and used for policy assessment, are IFM-CAP [43], which links emissions factors directly to the more granular defined production activities, and FARMDYM, which incorporates detailed emission accounting for different GHGs [44].

However, one critical aspect of PMP is an estimation of the explicit variable costs per crop with only the total variable costs per farm available. The generalized least square (LS) method, used in this study to estimate the cost function, enables calibration by overcoming the criticisms of Paris's three-step approach. This LS method, based on two steps, has the advantage of avoiding the unsolved problems of the arbitrary use of support values needed in the Maximum Entropy procedure [45,46], while using econometrics to correctly estimate the cost function, even in absence of exogenous accounting costs. The cost function, estimated in this way, makes it possible to differentiate the total variable costs of each crop between the explicit and implicit costs, relating to the agent's choice of what to produce and how. The calibration phase is followed by the simulation phase, which reproduces farmer behaviour triggered by new market and agricultural policy scenarios. The possibility of estimating an unambiguous cost function for each agent makes it possible to show a representation of the farm heterogeneity.

The PMP approach, developed according to the seminal work of Paris [47] and revised using the generalized LS method [48], introduces the following elements:

- farmer heterogeneity, made possible by an individual cost function for each farm in the sample;
- calibration performed for each farm, reproducing its observed activities using "self-selection". It can also reproduce an agent's "willingness" to adopt these activities that satisfy their family strategy, while being aware of alternative available processes;
- the exchange of resources (land, labour, and water, etc.) between agents made possible by links between farms;
- technology transfer between agents simulated by using the common cost function matrix, which, in the event of changes in market or policy scenarios, provides farmers with the economic and technological information related to those activities not included in their production plan, but which could be added or could replace the existing one(s).

### 2.2. The Model Structure

To simulate the effects on farmers' gross margins and structural changes caused by the introduction of carbon taxes, as well as other environmental constraints, agents (farm holders) are initialized with socio-economic characteristics (e.g., farmer age and family composition) and farm structure. The model is run in GAMS [49], with a two-stage structure: the calibration phase, which represents the "positive" component, and the simulation phase, which represents the "normative" component of the model.

Figure 1 represents the general structure of the model.

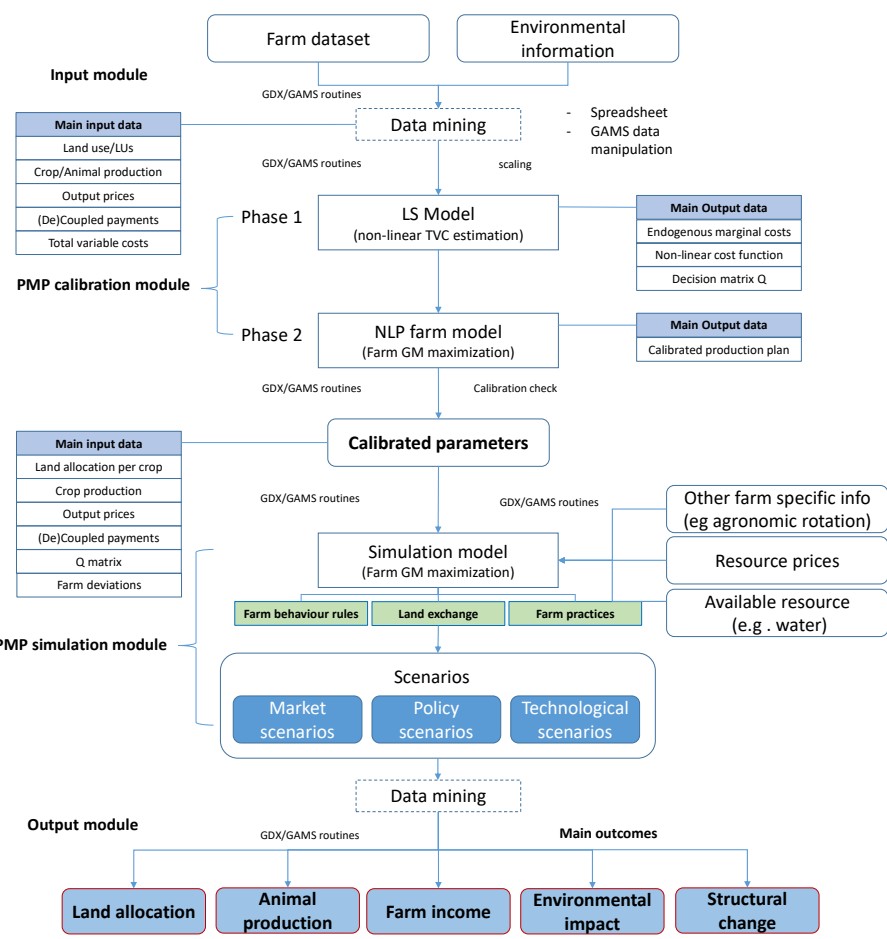

**Figure 1.** Model structure. Source: authors' own elaboration.

The calibration phase is performed using the LS technique on a sample of N farms. For each farm information on the production plan, prices and technical coefficients (the quantity of factors used to obtain one unit of product) are known. Only one limiting factor $b_n$, the available land at the farm level, ($n = 1, \ldots, N$), is considered. Unlike in other ABMs based on the PMP (e.g., SWISSland), the **Q** matrix composing the quadratic cost function is a full symmetric positive semi-definite matrix, ensured through Cholesky factorization, where **L** is a unit lower triangular matrix, **L**′ is its transpose, and **D** is a diagonal matrix whose elements are non-negative.

$$Q = LDL' \tag{1}$$

The coefficients, estimated in the quadratic cost function, provide flexibility to the model's responses towards farm simulations and information on the substitution and complementarity between agricultural activities [50].

AGRISP integrates the first and second phase of the standard PMP approach using the dual properties of PMP, avoiding the explicit inclusion of calibrating constraints. Moreover, this method makes it possible to fill the data gap of the Farm Accountancy Data Network (FADN). FADN is the most widely used database of agricultural information, but it only provides the total variable cost of the farm, without providing data on the variable costs per activity (**c**) [51]. The problem of implementing a PMP model without knowing **c** relates to the fact that the calibration constraints generate at least one associated shadow value equal to zero; otherwise, the shadow price for the structural constraint (land) will be equal to zero and an observed activity will be omitted in the Q matrix [50]. In the simulation

phase, the model maximizes a farm's gross margins using the quadratic cost function (**Q**) estimated in the calibration phase. The model, therefore, appears as follows:

$$\max_{x \geq 0} GM = p'x - \left\{ \frac{1}{2}x'\hat{Q}x + \hat{u}'x \right\} - (tx)'e \tag{2}$$

subject to:

$$Ax \leq b \tag{3}$$

where the unknown levels of production for each farm are indicated by the vector **x**, the output market prices are represented by the vector **p**, **Q** is the symmetric positive semi-definite matrix, and **u** is the vector of the marginal cost deviations per farm; **A** is the matrix of the technical coefficients and **b** is the vector of resources (land). The **Q** matrix does not represent the technology itself, but rather the technology costs related to the production choices; **t** and **e** represent, respectively, the $CO_2$ taxation and the vector of the emission factors.

Modelling dairy production relies on two main assumptions: the milk output price covers the costs of milk production (e.g., forage crop production costs, extra feed purchase costs, and cow maintenance costs, etc.), so that the milk price is greater than or equal to the milk accounting unit cost; and the livestock is closely linked to the available land, through the use of fodder crops produced on the farm. This is possible by adding Equation (4) to the simulation:

$$y_{nr}x_{n,milk} - x_{nr} \leq 0 \forall n \forall r \tag{4}$$

where $y_{nr}$ is the parameter of the feed requirement per unit of milk for the farm **n** ($n = 1, \dots, N$) and each fodder crop **r** ($r = 1, \dots, R$); $x_{n,milk}$ is the variable associated with the production of milk on the $n^{th}$ farm, and $x_{nr}$ is the variable for the production of fodder crops. Each farm reemploys all its forage produced to feed the dairy cows. This means that the market price of its fodder crops must be equal to 0, as the farm holder is not selling them. In this case, fodder includes meadows and pastures, alfa-alfa, forage maize, and other forages.

As noted above, farms can exchange land according to specific agent-based constraints that trace a one-to-one relationship between all the farms included in the sample, in the sense that each farm has the option to rent or rent out land with the other farms located in the neighbourhood. This relationship is represented in Equation (5):

$$\sum_j \left( A_{nj}x_{nj} \right) \leq b_n + Z_n - V_n \forall n. \tag{5}$$

Constraint (5) requires that the total land allocated to the different crops **j** ($j = 1, \dots, J$), $\sum_j \left( A_{nj}x_{nj} \right)$, ($j = 1, \dots, J$) must be less than or equal to the observed total available land at the farm level, $b_n$, plus the land rented ($Z_n$) minus the land leased ($V_n$). The land-exchange rules are designed based on the socio-economic characteristics of the farmers and on the assumption that the land price is the same for every farm and it is exogenously defined.

The land rented is represented as:

$$Z_n = \sum_m ZZ_{nm} \forall n \tag{6}$$

and the land rented out is represented as:

$$V_m = \sum_n VV_{nm} \forall m \tag{7}$$

where $ZZ_{nm}$ and $VV_{nm}$ are the matrix tracing the transfer of land for each pair of farms for renting and renting out, respectively. Furthermore, for each pair of farms, the land rented by one farm must be equal to the land leased by the other, as follows:

$$ZZ_{nm} - VV_{nm} = 0 \ \forall n \forall m. \tag{8}$$

To avoid a given farm renting and renting out land at the same time, a specific constraint has been added:

$$Z_n V_n = 0 \ \forall n \tag{9}$$

Finally, to ensure that the exchange of land is consistent with the total available land at the regional level, we establish that the total land rented must be equal to the total land rented out:

$$\sum_n Z_n - \sum_n V_n = 0 \tag{10}$$

Therefore, we assume that the exchange of land is limited to the farms located in the same agrarian region.

The behavioural rules implemented in AGRSIP and based on the social profile of the farmer defined for this study simulate that farmers older than 65 years of age with no successors are unlikely to rent additional land [29]. Renting out the entire property is assimilated to farm withdrawal.

### 2.3. Sample Analysis

The sample investigated is limited to farms located in the Emilia-Romagna NUTS-2 region. It refers to the 2020 Italian FADN (RICA) observation that counts 710 farms. The dataset includes information on their geographical location (region, province, altitude, and agrarian zone), agricultural practices (conventional or organic), household characteristics (age and gender of the farm holder and number of potential farm holder's successors), land use, specific production costs per crop (cost of seeds, fertilizers, pesticides, energy, and water), gross total product, and CAP payments. The "agrarian region" spatial definition is a peculiarity of the RICA and further segments Italian provinces (NUTS3) based on their geographical location and altitude range. In Emilia Romagna 69% of the UAA is located in flatland (Figure 2).

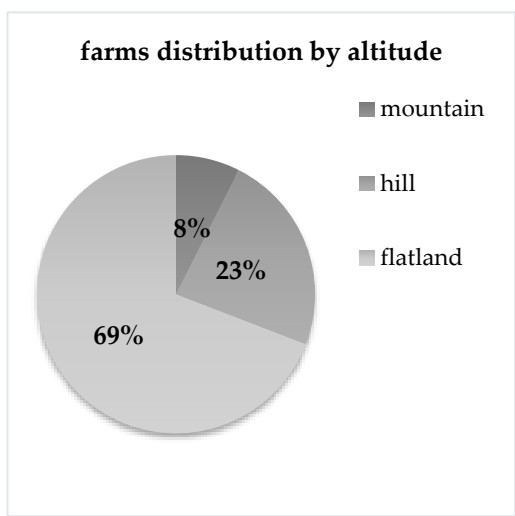

**Figure 2.** Farms distribution by altitude. Source: Emilia-Romagna 2020 FADN.

The technical–economic orientation and number of farms associated is shown in Table 1, which also reports the weighted number of farms, as RICA provides a sample weight for each farm to be representative of the whole universe. Farms with data inconsistencies, as well as those with more than 1000 ha, have been removed from the sample, as they are not statistically representative.

**Table 1.** Number of farms in the Emilia-Romagna 2020 FADN by type of farming.

| Farm Technical Orientation | Type of Farming | Sample | Weighted Sample |
|---|---|---|---|
| Arable crops | 1 | 310 | 15,351 |
| Horticulture | 2 | 8 | 411 |
| Permanent crops | 3 | 160 | 9084 |
| Dairy cattle | 450 | 91 | 3306 |
| Other herbivores | 460; 470; 481; 482; 484 | 24 | 2308 |
| Granivores | 5 | 30 | 677 |
| Polyculture | 6 | 67 | 3371 |
| Mixed farming | 7 | 2 | 33 |
| Mixed (crop–livestock) | 8 | 18 | 919 |
| Total | | 710 | 35,459 |

Source: Emilia-Romagna 2020 FADN.

Table 2 shows the composition of the sample in terms of farm structure and holder's age, distinguishing between farms specialized in dairy cattle (dairy farms) and farms with other technical orientations (other farms). The FADN sample of Emilia-Romagna is characterized by a prevalence of farms smaller than 10 ha (44.8%). In terms of the holder's age and technical orientation, the largest categories are non-dairy farms with farm holders aged 41–64 (44.1%) and 65 or above (41.9%). Young farm holders account for only 5.8%. Most of the farms are located in flat lands (69%) (Figure 1).

**Table 2.** Emilia-Romagna 2020 FADN sample composition according to holder's age, type of farming and farm dimension.

| Technical Orientation | Dairy Farms | | | Other Farms | | | Total | % |
|---|---|---|---|---|---|---|---|---|
| Holder's Age | ≤40 | 41–64 | ≥65 | ≤40 | 41–64 | ≥65 | | |
| <10 ha | | 289 | 68 | 593 | 7235 | 7696 | 15,879 | 44.8 |
| 10–20 ha | 191 | 189 | 153 | 444 | 3248 | 4153 | 8379 | 23.6 |
| 20–50 ha | 121 | 850 | 326 | 295 | 3161 | 2365 | 7120 | 20.1 |
| 50–100 ha | 84 | 304 | 372 | 187 | 1404 | 380 | 2730 | 7.7 |
| 100–300 | 30 | 200 | 129 | 93 | 496 | 247 | 1196 | 3.4 |
| >300 ha | 0 | 0 | 0 | 31 | 109 | 15 | 155 | 0.4 |
| Total | 426 | 1831 | 1049 | 1644 | 15,654 | 14,856 | 35,459 | |
| % | 1.2 | 5.2 | 3.0 | 4.6 | 44.1 | 41.9 | | |

Source: Emilia-Romagna 2020 FADN.

Considering the livestock sector, Emilia-Romagna accounts for 11.4% of the livestock units bred in Italy; in more detail: 10% bovines, 12% swine, and 18% poultry, representing 15.2% of the national animal production value (2357.3 million EUR) [6]. Emilia-Romagna also produces 16% of Italian milk: in 2020, cow milk production stood at 2,029,257 tonnes, placing the region in second place for milk production, after Lombardia (44%).

Cheese production is strongly rooted in the region: in 2020, 89.2% of the milk produced in the area between the Po and the Reno rivers was allocated to the production of 140,000 tonnes of Parmigiano Reggiano, and 325,700 tonnes of regional milk (0.016%) was used to produce 24,000 tonnes of Grana Padano cheese [52].

Consequently, due to its strong agro-industrial vocation, the ER region is responsible for 10.4% of Italian livestock-related GHG emissions (2059 thousand tonnes) and for 9% of the national ammonia emissions (23,114.8 tonnes of $NH_3$) [7]. The amount of $CO_2$eq emitted per tonnes of crop, shown in Table 3, is calculated on the basis of the estimated emissions per hectare, or per livestock unit (LSU), using the ICAAI methodology (Impronta Carbonica dell'Azienda Agricola Italiana) developed by CREA-PB on the basis of the IPCC guidelines for establishing a national inventory of greenhouse gas emissions [53–55].

**Table 3.** Carbon footprint expressed in tonnes of $CO_2$ equivalent per hectare.

| Aggregated Crops | Crops | T $CO_2$eq/Ha |
|---|---|---|
| Cereals | durum wheat | 1.66 |
| | soft wheat | 1.55 |
| | sorghum | 1.33 |
| | rice | 8.50 |
| | barley | 1.33 |
| | other cereals | 1.33 |
| Maize | maize | 3.52 |
| Forages | other forages | 0.67 |
| | alfa alfa | 0.50 |
| | forage maize | 1.77 |
| Proteic/Oilseeds | oilseeds | 0.82 |
| | sunflower | 0.82 |
| | soja | 0.81 |
| | protein crops | 1.04 |
| Industrial Crops | potato | 2.27 |
| | industrial tomato | 2.11 |
| | beetroot | 1.45 |
| Meadows and Pastures | meadows and pastures | 2.24 |
| Dairy Milk | milk | 5.50 |

Source: ICAAI methodology of CREA-PB.

Besides carbon emissions and nitrogen production, water consumption is also evaluated using the water footprint data (Table 4) calculated by Hoekstra and Mekonnen [56], as the sum of:

- green water, which is water naturally embedded in the rhizosphere and available for plant assimilation;
- blue water, which is surface water or groundwater for irrigation;
- grey water, which is the volume of water necessary to dilute ecotoxic compounds to restore specific quality standards.

**Table 4.** Water footprint expressed in m$^3$ of water per hectare.

| Water Footprint (m$^3$/Ha) | Crops | Green Water | Blue Water | Grey Water |
|---|---|---|---|---|
| Cereals | durum wheat | 996.7 | 23.2 | 172.2 |
| | soft wheat | 996.7 | 23.2 | 172.2 |
| | sorghum | 602.6 | | 228.2 |
| | rice | 511.5 | 562.5 | |
| | barley | 996.7 | 23.2 | 172.2 |
| | other cereals | 3676.1 | 210.3 | 1293.3 |
| Maize | maize | 368.2 | 158.8 | 160.4 |
| Forages | other forages | 8021.7 | 641.7 | 697.2 |
| | alfa alfa | 8021.7 | 641.7 | 697.2 |
| | forage maize | 368.2 | 158.8 | 160.4 |
| Proteic/Oilseeds | oilseeds | 10,829.8 | 685.7 | 13.3 |
| | sunflower | 1595.1 | 130.9 | 310.8 |
| | soja | 1115.7 | 221.8 | 2.9 |
| | protein crops | 7898.6 | | 97.2 |

**Table 4.** *Cont.*

| Water Footprint (m³/Ha) | Crops | Green Water | Blue Water | Grey Water |
|---|---|---|---|---|
| Industrial Crops | potato | 371.7 | 86.9 | 31.5 |
| | industrial tomato | 58.9 | 32.9 | 11.9 |
| | beetroot | 100.0 | 13.0 | 18.0 |
| Meadows and Pastures | meadows and pastures | 8021.7 | 449.2 | 488.0 |
| Dairy Milk | milk | 626.0 | 77.0 | 102.0 |

Source: Water Footprint Network.

*2.4. Scenarios*

To assess, ex-ante, the impact of the introduction of an increasing carbon tax and quotas on the nitrogen from bovine manure, as well as farmers' responses in changing their production plans and resource allocation, six scenarios are modelled in AGRISP, in addition to the situation observed at the time of calibration, represented as "s_cal".

The first scenario, "s_land", defines the possibility of renting or renting out arable land as a way of making optimal use of farm resources. Farmers can adopt structural strategies, renting out all their land and abandoning the market or renting out just a part of their land and continuing farming. The exchange land scenario, "s_land", highlights how some farmers opt for a more efficient combination of their land, increasing their size while others leave the sector. The cost for renting a hectare of arable land was set at 589 EUR by the Land Market Research of CREA-PB for 2020 [57]. The "s_land" scenario is considered as the baseline for comparison with the following scenarios.

The "s_nitrogen" scenario simulates the right to spread manure according to the EU Nitrate Directive 91/676/CEE, which is aimed at reducing and preventing nitrate water pollution from agricultural sources [58]. The Nitrate Directive requires Member States to be responsible for identifying pollution sources, for designating "Nitrate Vulnerable Zones" (NVZs), and for designing appropriate action programs.

In Emilia-Romagna, the European Nitrate Directive is transposed into the Regional Regulation 15/12/2017 No.3, which identifies NVZs exclusively in flatlands, representing 29.6% of the regional agricultural flatland. The Regulation sets the limit that 170 kg of nitrogen from livestock manure can be spread annually over one hectare in an NVZ and there is a limit of 340 kg in other areas [59]. According to Specific Objective no.5 "Promoting sustainable development and efficient management of natural resources such as water, soil and air", [60], Emilia-Romagna NVZs account, respectively, for 60.4%, 32.7%, and 13.3% of UAAs in flatlands, hills, and mountain zones. Therefore, we apply the maximum amount of spreadable nitrogen for each altimetric zone according to the percentage of NVZs. To do so, we therefore consider the quantity of the nitrogen produced by livestock for each farm, knowing the number of cows used for milk production, with emissions of 82.8 kg of $NO_2$ per dairy cow (0.42% nitrogen content of dairy cow manure) and 36 kg of $NO_2$ per rebreeding cow, according to the Regional Regulation noted above. Farm holders under the constraint of 170 or 340 kg of N/ha, and on the basis of the quantity of nitrogen produced by their livestock and their Utilized Agricultural Area (UAA), can decide to either reduce their number of cows or rent more land from non-livestock farms, in order to spread the exceeding manure. Dairy farmers can acquire rights to pollute and spread the exceeding manure from non-livestock farms that need nitrogen fertilizers, paying a cost of 150 EUR/ha plus 69 EUR/t nitrogen for the spreading cost. This cost is calculated based on the average of 40.63 tons of manure per hectare at a price of 80 EUR/hour and the capacity (45 tonnes of manure) of a big manure tank. Transportation costs are not considered, as the farms are not geolocated in the FADN sample, but it is assumed that only neighbouring farms exchange rights to pollute, limiting these exchanges within an agrarian region. The reason why we consider this scenario is to estimate, a priori, based on the number of LSUs and the land endowment of the dairy farms, whether the Nitrogen Directive

in Emilia-Romagna can be respected or if dairy farmers must search for manure disposal solutions outside of their farm boundaries.

Four different taxes (20, 50, 100, and 150 EUR/tCO$_2$eq) are applied to measure how this additional cost influences farmers' production choices and gross margins. Each taxation level corresponds to a different scenario ("s_em20", "s_em50", "s_em100", and "s_em150").

All the above scenarios include the default policy measures of CAP 2023–2030, such as greening payments, single payments, and crop coupled payments, funded through Pillar I, under the 2014–2020 CAP reform [61]. It is also assumed that farmers over 65 with no successors receive a retirement pension of 1000 EUR/month.

## 3. Results

The impact of different renting strategies, shown in Table 5, explains the structural variation due to the introduction of a rule imposing the full utilisation of the available land. The total number of farms decreases from 35,459 in "s_cal" to 33,498 (−5.5%) in "s_land", with a bigger impact on non-dairy farms (−5.9%) than dairy farms (−2.0%). Considering the farm holders' age for the whole sample, the number of farms decreases by −7.6% in the age range of 41–65, by 5.8% in the range of ≤40, and by 3.2% in the range of ≥65.

**Table 5.** Variation of number of farms per class of holders' age under.

|  | Dairy Farms | | Other Farms | | All Farms | |
| --- | --- | --- | --- | --- | --- | --- |
| **Holder's Age** | **s_cal** | **s_land** | **s_cal** | **s_land** | **s_cal** | **s_land** |
| **≤40** | 426 | 426 | 1644 | 1523 | 2070 | 1949 |
| **41–64** | 1831 | 1763 | 15,654 | 14,391 | 17,485 | 16,155 |
| **≥65** | 1049 | 1049 | 14,856 | 14,346 | 15,904 | 15,395 |
| tot | 3306 | 3238 | 32,153 | 30,260 | 35,459 | 33,498 |

Source: authors' own elaboration.

The "s_land" scenario, used as baseline for the "s_nitrogen" and increasing emissions taxation scenarios, depicts how some farmers opt for structural adaptation strategies to find new forms of economic efficiency, whereas other farmers decide to leave the sector when pushed to fully use their land endowment.

Figure 3 depicts the effect of environmental measures on the baseline scenario. The decrease in the number of farms is very limited in absolute numbers: it is more evident for non-dairy farms, with minus 502 farms, while in the dairy sector, only 98 farms decide to abandon their activities when a higher CO$_2$ tax ("s_em150") is introduced.

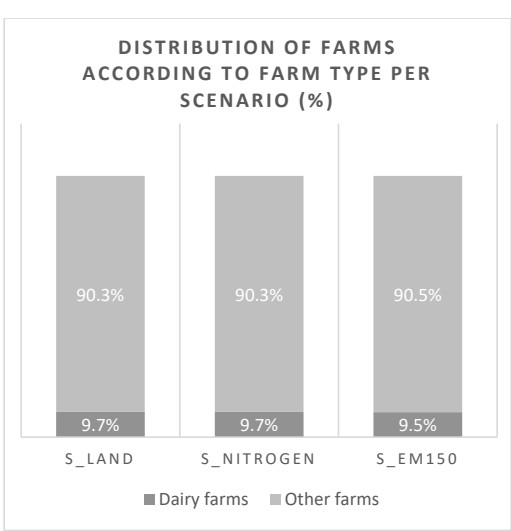

**Figure 3.** Percentage distribution of farms according to farm type in different scenarios. Source: authors' own elaboration.

Considering the farm holders' age (Table 6), dairy farms decrease in the range of 41–64, while non-dairy farms decrease in the range of ≥65. The introduction of the Nitrate Directive ("s_nitrogen") has no influence on the number of farms.

**Table 6.** Number of farms per policy scenario and class of holder's age.

| Dairy Farms | s_land | s_nitrogen | s_em20 | s_em50 | s_em100 | s_em150 |
|---|---|---|---|---|---|---|
| ≤40 | 426 | 426 | 426 | 426 | 426 | 426 |
| 41–64 | 1763 | 1763 | 1763 | 1763 | 1733 | 1665 |
| ≥65 | 1049 | 1049 | 1049 | 1049 | 1049 | 1049 |
| tot | 3238 | 3238 | 3238 | 3238 | 3208 | 3140 |
| **Other farms** | **s_land** | **s_nitrogen** | **s_em20** | **s_em50** | **s_em100** | **s_em150** |
| ≤40 | 1523 | 1523 | 1506 | 1505 | 1505 | 1506 |
| 41–64 | 14,391 | 14,391 | 14,436 | 14,405 | 14,361 | 14,201 |
| ≥65 | 14,346 | 14,346 | 14,291 | 14,248 | 14,164 | 14,052 |
| tot | 30,260 | 30,260 | 30,234 | 30,159 | 30,031 | 29,759 |
| **All farms** | **33,498** | **33,498** | **33,472** | **33,397** | **33,239** | **32,899** |

Source: authors' own elaboration.

Figure 4 shows the impact of policy scenarios on farm gross margins (GM), aggregated at the regional level (Table A1). The GM decreases slightly in the scenario "s_nitrogen", but this reduction is substantial and increases along with a tax increase (from −4.3% with a tax of 20 EUR/tCO$_2$eq, up to −24.7% in "s_em150").

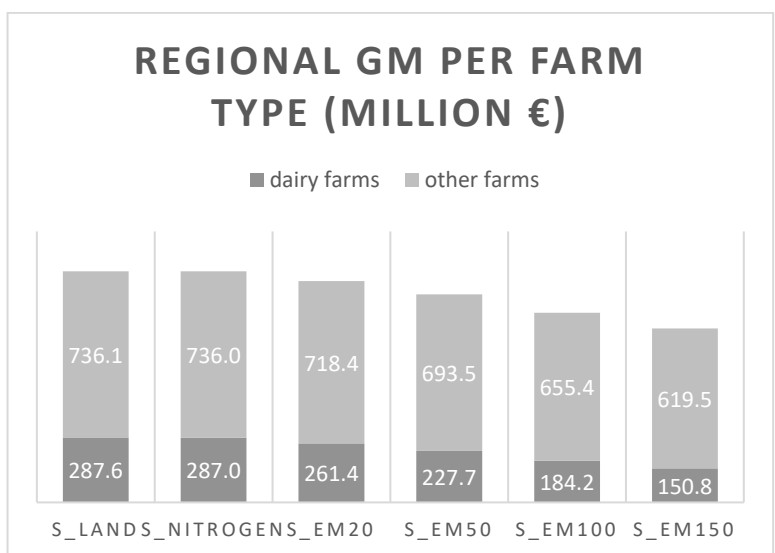

**Figure 4.** Regional gross margin per farm type, expressed in million EUR. Source: authors' own elaboration.

Looking at the farms' technical orientations (Figure 5), the average gross margins of dairy farms are more affected by both the spreading constraints and the carbon taxes compared to the gross margins of non-dairy farms (Table A2). The "s_nitrogen" scenario only affects dairy farms, with a reduction of −0.2% in the farms' gross margins, while the carbon tax scenarios have an economic impact on all farms, but with a stronger effect on dairy farms (−45.9% against −14.4% of non-dairy farms).

Considering the economic impact of the different measures on farm income by farm holders' age, livestock holdings are those that show the greatest reduction in farm GM due to the introduction of CO$_2$ taxation (Tables A3 and A4). Taxation also heavily impacts older farmers, for whom the state pension does not seem to contribute to the further existence of these farms.

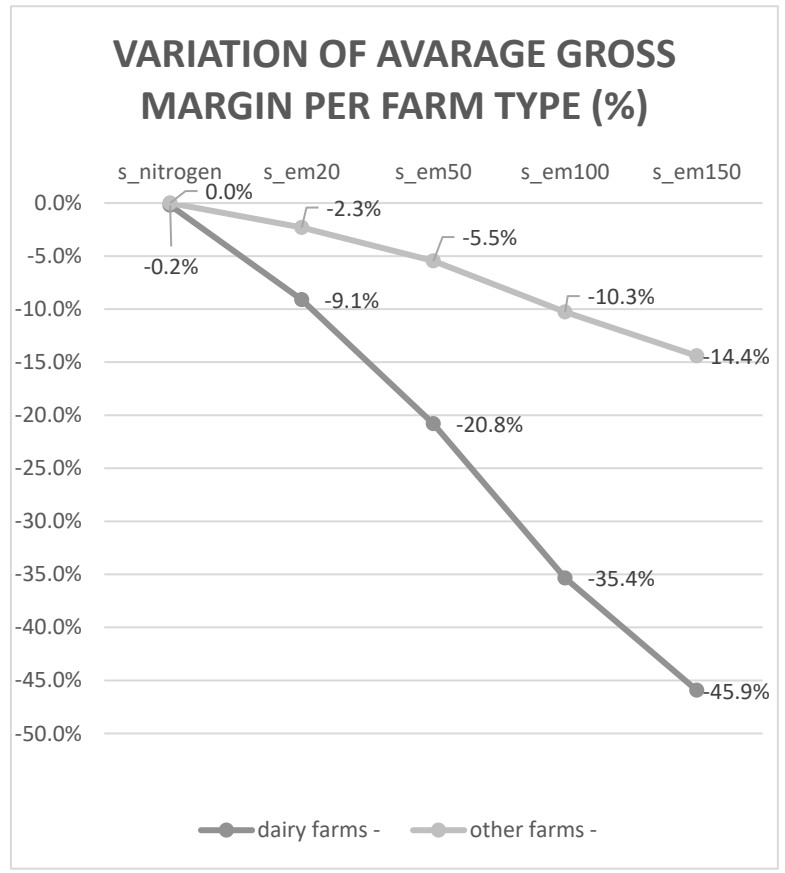

**Figure 5.** Average farm gross margin per farm type, expressed in EUR. Source: authors' own elaboration.

The introduction of $CO_2$ taxation also has a significant effect on farm production organisation by modifying land use (Tables 7 and 8). The "s_nitrogen" scenario, while not affecting the number of farms, modifies this production organisation by reducing the area allocated to silage for cows outside the Parmigiano Reggiano PDO (Protected Designation of Origin) area. On the other hand, the scenarios of rising taxation push farms towards a more extensive cultivation of crops with less environmental impact. In fact, the most heavily penalised crops are maize and industrial crops (tomato, potato, and beetroot), which are reduced by 65.4% and 56.7%, respectively. These crops would be replaced by fodder crops (alfa-alfa, forage maize, and other forages, +17.7%), protein crops, and oilseeds (+6.6%) and set-aside (+21.7%).

**Table 7.** Land allocation in thousands of hectares by crop type.

| UAA (1000 ha) | s_land | s_nitrogen | s_em20 | s_em50 | s_em100 | s_em150 |
|---|---|---|---|---|---|---|
| Cereals | 188.1 | 187.2 | 188.5 | 187.6 | 189.5 | 191.1 |
| Forages | 380.9 | 382.6 | 412.2 | 427.6 | 440.5 | 448.5 |
| Maize | 45.5 | 45.1 | 38.9 | 31.0 | 21.9 | 15.7 |
| Proteic/Oilseeds | 72.1 | 73.5 | 76.8 | 77.4 | 77.3 | 76.9 |
| Meadows Pastures | 68.1 | 68.1 | 67.1 | 65.9 | 65.9 | 65.9 |
| Industrial crops | 77.3 | 75.6 | 48.7 | 42.4 | 36.8 | 33.5 |
| Greening | 2.5 | 2.5 | 2.5 | 2.6 | 2.7 | 3.0 |
| Total | 834.6 | 834.6 | 834.6 | 834.6 | 834.6 | 834.6 |

Source: authors' own elaboration.

**Table 8.** Percentage variation in land allocation compared to s_land.

| % Variation | s_nitrogen | s_em20 | s_em50 | s_em100 | s_em150 |
|---|---|---|---|---|---|
| Cereals | −0.5 | 0.2 | −0.3 | 0.7 | 1.6 |
| Forages | 0.4 | 8.2 | 12.2 | 15.6 | 17.7 |
| Maize | −0.8 | −14.5 | −31.8 | −51.9 | −65.4 |
| Proteic/Oilseeds | 1.9 | 6.4 | 7.3 | 7.1 | 6.6 |
| Meadows Pastures | −0.1 | −1.6 | −3.2 | −3.3 | −3.3 |
| Industrial crops | −2.2 | −37.0 | −45.1 | −52.4 | −56.7 |
| Greening | 0.0 | 1.1 | 6.2 | 11.3 | 21.7 |

Source: authors' own elaboration.

The increase in fodder crops (alfalfa, fodder maize, and other fodder crops) is due to non-dairy farms selling these products on the market, while dairy farms use them as cow feed. Meadows and pastures decrease for dairy farms in the "s_em50" scenario, then increase again. The meadows and pastures of non-dairy farms decrease progressively as the tax increases (Table 9). It must be highlighted that, in AGRISP, dairy farms do not buy fodder from other dairy farms, as they only use fodder produced on their farm. On the other hand, other farms sell fodder on the market at the current FADN price. The increase in grazing meadows is justified by the farmers' strategy of moving towards crops that emit less $CO_2$ in a strategy of progressive extensification.

**Table 9.** Land allocation of reused crops (forages and meadows and pastures) according to the farm type.

| Forages (ha) | s_land | s_nitrogen | s_em20 | s_em50 | s_em100 | s_em150 |
|---|---|---|---|---|---|---|
| **Dairy farms** | 83,439.7 | 84,633.7 | 83,396.3 | 85,807.4 | 86,966.8 | 87,850.4 |
| % variation | - | 1.4 | −0.1 | 2.8 | 4.2 | 5.3 |
| **Other farms** | 297,504.4 | 297,960.5 | 328,840.5 | 341,747.3 | 353,490.2 | 360,636.7 |
| %variation | - | 0.2 | 10.5 | 14.9 | 18.8 | 21.2 |
| **Meadows Pastures (ha)** | s_land | s_nitrogen | s_em20 | s_em50 | s_em100 | s_em150 |
| **Dairy farms** | 22,071.5 | 21,783.9 | 20,667.7 | 20,208.1 | 21,163.1 | 21,472.0 |
| % variation | - | −1.3 | −6.4 | −8.4 | −4.1 | −2.7 |
| *Other farms* | 46,073.6 | 46,283.9 | 46,393.0 | 45,728.7 | 44,751.4 | 44,412.9 |
| % variation | - | 0.5 | 0.7 | −0.7 | −2.9 | −3.6 |

Source: authors' own elaboration.

All the policy scenarios introduced show a decrease in the number of dairy cows (Figure 6, Table A5). This decrease is due to the introduction of nitrogen pollution quotas in "s_nitrogen", and to the introduction of $CO_2$ taxes that impact the GHG emissions associated with milk production.

Along with a reduction in the number of dairy cows (Figure 7, Table A6), the nitrogen emissions coming from manure, expressed in tonnes of N, are also reduced. The emission estimation takes into account the nitrogen produced by dairy cows and breeding cows.

Considering carbon emissions, the application of a nitrogen scenario generates a reduction of −1.2% in $CO_2$ emissions (Figure 8). The most impacted products are dairy milk (−1.8%) and industrial crops (−3.8%). Emissions related to protein crops and oilseeds increase by 2.6%. The $CO_2$ tax scenarios also generate overall reductions in $CO_2$ emissions of, respectively, −9.4%, −21.7%, −35.7%, and −44.6%. Decreases occurred for dairy products, maize, industrial crops, cereals, and meadows and pastures. Forages and protein/oilseeds emissions increase (Tables A7 and A8).

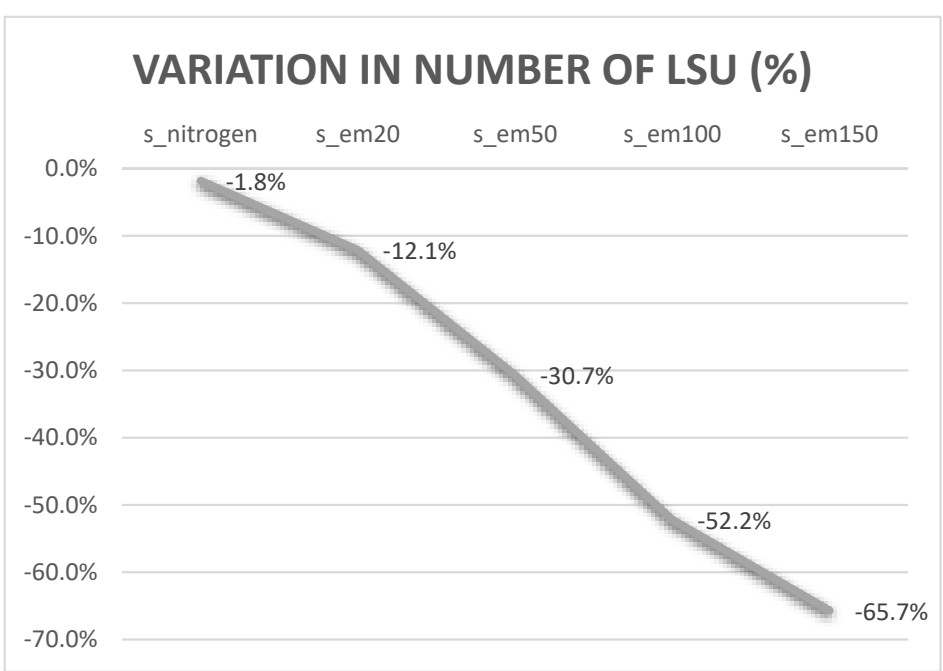

**Figure 6.** Percentage variation in the number of livestock per scenario. Source: authors' own elaboration.

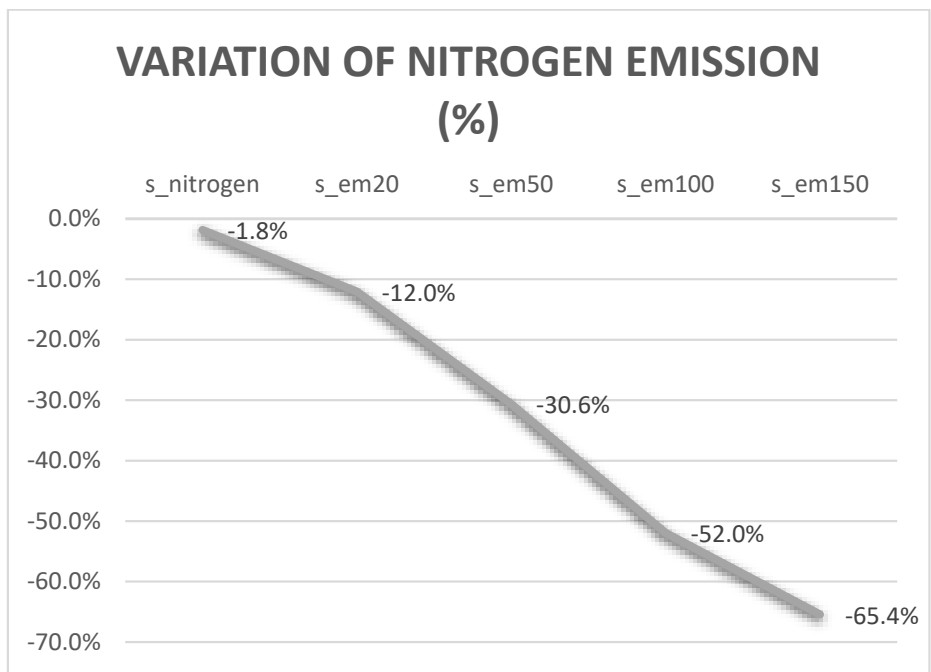

**Figure 7.** Percentage variation in nitrogen emission per scenario. Source: authors' own elaboration.

Finally, the model assesses water consumption (Figure 9). The shift of allocation plans towards less carbon-emitting activities described above leads to an increase in the total water consumption, since these crops require more water. The biggest increase (+10.5%) is found in scenario "s_em150", mainly due to an increase in the water footprint of fodder crops (Tables A9 and A10).

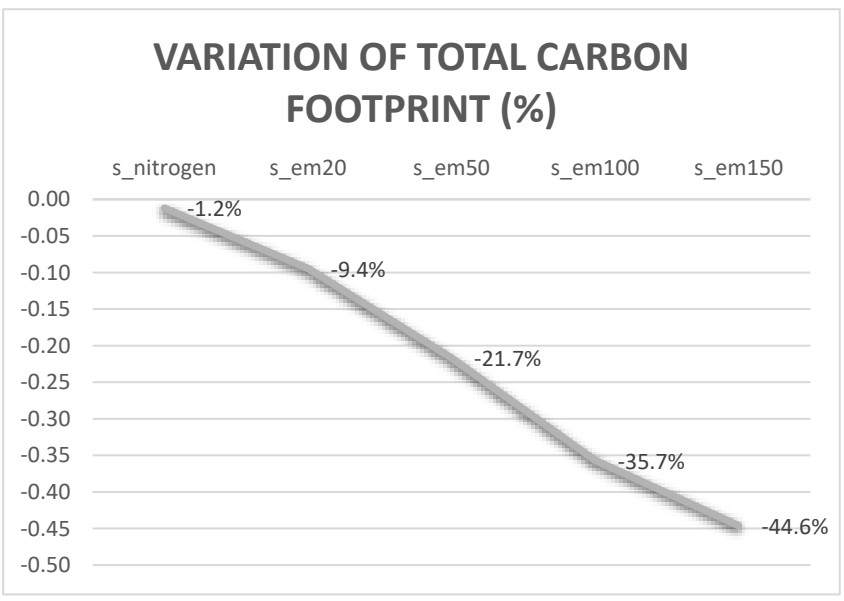

**Figure 8.** Percentage variation of total carbon footprint. Source: authors' own elaboration.

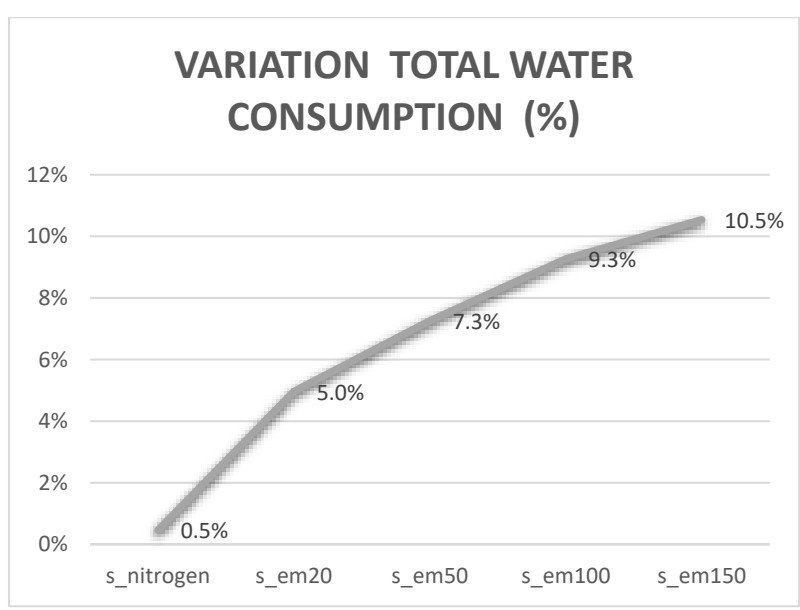

**Figure 9.** Percentage variation of total water footprint. Source: authors' own elaboration.

## 4. Discussion

This article presented an Agent-Based bio-economic farm model with the aim of assessing the structural, production, environmental, and economic impact of an increasing tax on climate change gas emissions related to milk production under the current CAP payment system. The analysis was performed by linking the properties of PMP to agent-based modelling and by using the FADN farm dataset for the Emilia-Romagna region (Italy). The elements that characterise the ABM are:

- the development of one model for each farm belonging to the sample, regardless of the farm type, including dairy farms;
- the calibration performed on the observed production data;
- the possibility for farms to exchange technologies and activities;
- the possibility for farmers to exchange fixed factors, such as land;
- the inclusion of behavioural constraints that regulate the strategy of each farm holder.

The model did not simulate the soil–plant interaction. The data on water consumption were taken from the FADN, and the C and N balances of various types of crops were not included in the calculation. The yields calculated in the FADN were used to build the substitution matrix and the agricultural practices were not considered in detail.

The variable cost per crop, needed to build the Q matrix, was, in this work, estimated through a generalised least square approach. Even though these costs are available in the Italian FADN, we decided to opt for this methodological approach to compare the results with real data and define a method adoptable for other European regions where variable costs are not reported.

In context of the current CAP, the Dairy Farm Agent-Based Model AGRISP made it possible to simulate and assess the impact of environmental policy scenarios for a heterogeneous farm population according to the farm type, structure, and age of the farmers. Six policy scenarios from Pillar I were designed to optimize the nitrogen distribution on the soil and the reduction in $CO_2$. Measures of Pillar II were not included in this work.

Yet, the model only considered the dairy cow compartment, without considering the impact of other livestock, especially pig and poultry production, that still have a substantial impact on GHGs. The assumption that dairy cow livestock is closely linked to the available land, using the fodder crops produced on a farm, is consistent with usual farm practice, according to which, all fodder production obtained on the farmland is addressed to feed livestock and with the rules imposed by the Parmigiano Reggiano PDO Code of Specification [62]. The Code of Specification states that at least 50% of the dry matter of the fodder used must be produced on the farmland, and at least 75% of the dry matter of the fodder must be produced within the Parmigiano Reggiano cheese production area. The strict linkage between the feed requirement and farmland allows for the imposition of an implicit structural limit on the animal production capacity at the farm level and the merging of land cultivation and livestock in one single farmer's optimisation strategy. In other words, for dairy farms, the hypothesis is to maximise the value of the milk production intended for a transformation from fodder to milk. In this context, the dairy farm is pushed to first employ all the fodder produced in-farm until the economic equilibrium condition is fulfilled: the marginal cost of milk production, represented by the fodder (and feed) marginal cost, is equal to the marginal revenue (milk price). It is worth noting that concentrates and off-farm hay procurement were not missing in the model, but they were included as a component of the milk production cost. Therefore, one of the main characteristics of the livestock PMP approach in AGRISP was the absence of the fodder consumption function based on technical coefficients, which was replaced by the cost function.

The results can be interpreted according to the specific environmental, structural, economic, and social policy lever considered. These aspects are all interlinked, as farms change their production orientation and structure, significantly affecting the overall farm income. Despite a decrease in the number of LSUs, the increase in the land allocation of reused crops (forages and meadows and pastures) was due to the enforced agronomic constraint of equality between the total production of forage (including alfa-alfa, soja, and protein crops, plus some industrial crops such as sugar beet and tomatoes) and the total production of cereal (wheat, maize, barley, and sorghum).

The nitrogen scenario did not seem to have a big impact on any of the indicators analysed, however, the model proved to be suitable to simulate further restrictions that could be introduced into the regulation, given the persistence of the nitrogen issue.

The repercussion of $CO_2$ taxation on farms' structures and rural regions, revealed through the use of agent-based models, was significant. If output prices were assumed to remain unvaried, the introduction of a $CO_2$ tax would have a greater impact on the most polluting processes, intensive crops, and dairy cows, and the more intensive farms that would then opt for new production strategies to become more environmentally sustainable. Increased environmental sustainability can be achieved by reducing soil pressure thanks to the presence of fewer animals per hectare, the use of more sustainable fodder, and the possibility of redistributing nitrate quotas to non-livestock farms.

It is interesting to note that, with the current CAP (2023–2030), the impact of greening measures on land is also low, as it was in the previous programming period [63]. However, an increase in fodder production, as it is a less intensive crop, triggers indirect positive effects on the environment and a consequent improvement in the environmental indicators covered by the Rural Development Plan (RDP), monitoring, namely: Farmland Bird index, High nature value farming, and the Soil organic matter in arable land [64]. The introduction of an increasing tax on $CO_2$ emissions would further ameliorate this positive effect. However, the possibility of exchanging land favours the most efficient farms, which increase their size to the detriment of inefficient farms, with consequent economic and social impacts.

This finding was a specific and valuable output of the Agent-Based Model, which considered the social characteristics of the farm holders, assuming their age as an indicator of the social renewal of the agricultural population. The age of farm holders is an important element in the definition of regional and national strategies aiming to increase the number of young farmers or to discourage farm abandonment [65,66]. In the case of Emilia-Romagna, the analysis showed how young farmers managing dairy farms are more resilient compared to other farm types, but a more closely tailored policy could reduce these drop-out rates further.

AGRISP, used in this work, represents a further advancement in agent-based modelling for agricultural policy analyses. Research attention is currently focused on developing this category of model, which is particularly well suited to representing farm strategies given their spatial production context and the possibility of interactions between farms [67]. Within the Agrimodel Cluster (https://agrimodels-cluster.eu/ (accessed on 14 June 2023) funded by Horizon2020, three projects are dedicated to assessing and developing ABMs for agricultural policy analyses: AGRICORE, BestMap, and MindStep. The model presented in this paper, delivered under the AGRICORE project, differs from the other models, as it uses PMP to estimate the cost functions per farm, allowing for the exchange of technologies, as well as other factors, between agents. The model can also be considered as "generalised", in the sense that it considers all the sampled farms regardless of their specialisation. Moreover, AGRISP does not make use of information exogenous to the farm, except for the price of the production factor exchanged, in this case, land. Finally, AGRISP is particularly well suited to simulating different agricultural and environmental policy instruments involving different forms of direct payments and subsidies. It is particularly appropriate for ex ante analyses aimed at evaluating the congruence of RDP with Farm to Fork and Green Deal objectives.

## 5. Conclusions

Simulating the state of the application of the nitrate directive and the introduction of a $CO_2$ tax in the agricultural sector to decrease GHG emissions and promote mitigation actions represents a helpful exercise for developing effective environmental policy measures at the farm level. The results of the simulations are meaningful for policy, in as much as the model can represent the farm heterogeneity characterizing the agricultural sector under investigation. The economic models usually adopted for assessing agricultural policy in the EU neglect the potential interactions among farms in exchanging limited resources, such as land, and simulate farm behaviour by aggregating information rather than using individual data. Aggregating FADN data by region or farm type simplifies the implementation of the mathematical programming methodology but loses essential information on the farming system (e.g., production plan) and farmer behaviour. AGRISP, as an agent- and micro-based model, exploits the farm data included in FADN, providing policymakers with clear, consistent, and understandable information on farmers' responses to new policy measures. Given the lack of variable costs per agricultural activity in the FADN database, AGRISP estimates this economic information by farm and agricultural activity in a two-step PMP approach.

This research work contributes to enriching the set of economic mathematical models adopted to evaluate the agri-environmental policies in the EU by proposing an agent-based model capable of recovering information on the exact order of choice of each farmer and simulating farm behaviour, thanks to the constraints imposing interactions among

farms and behavioural rules that mainly define the social features in the simulation phase. This model is distinguished from the other agent-based models aimed at assessing agri-environmental policies by the fact that:

- the two-step PMP approach estimates the economic complementarity and substitution relationships characterizing each agricultural activity;
- each farm can activate alternative agricultural activities according to the self-selection information embedded in the farm cost function;
- new agricultural activities and technologies can be accommodated in the model as options for the current production plan.

Our results showed that dairy farms are more sensitive to taxes on carbon emissions than other farms, both in terms of activity withdrawals and gross margin reduction. The climate mitigation strategy promoted by an increasing $CO_2$ tax would lead farmers to reduce their livestock and substitute more energy-intensive crops, such as maize and processed tomato, with cereals and fodder crops. Other studies on the greening of the CAP have confirmed that a more environmentally friendly CAP would depress internal animal production, increasing the dependency on non-EU markets. The experience of the introduction of green taxes in the EU indicates the need to gradually increase the tax level to allow for agents to adapt to this new policy mechanism. From a short-term perspective, as suggested by our model, a high tax on $CO_2$ emissions would worsen the viability in rural areas and the trade balance for milk. Further development of this research work could contemplate the integration of AGRISP with other complementary models, the introduction of other forms of production factor exchanges between agents, and the repayment of tax revenues in form of incentives for more sustainable agricultural practices to adjust the environmental–economical trade-off.

**Author Contributions:** Conceptualization, L.B. and M.D.; methodology, F.A. and M.D.; software, S.C.; validation, L.B., S.C. and M.D.; formal analysis, L.B.; investigation, L.B. and S.C.; resources, F.A.; data curation, L.B. and S.C.; writing—original draft preparation, L.B.; writing—review and editing, F.A. and S.C.; visualization, S.C.; supervision, L.B.; project administration, L.B.; funding acquisition, F.A. All authors have read and agreed to the published version of the manuscript.

**Funding:** This research was funded by EU Horizon 2020 AGRICORE, grant number 816078.

**Data Availability Statement:** Research uses data from the elaboration of original RICA data which are unavailable due to privacy prescription.

**Conflicts of Interest:** The authors declare no conflict of interest.

## Appendix A. Detailed Calculation and Tables

**Table A1.** Overall regional gross margin and gross margin per hectare.

| Regional GM (Million EUR) | s_land | s_nitrogen | s_em20 | s_em50 | s_em100 | s_em150 |
|---|---|---|---|---|---|---|
| **Dairy farms** | 287.6 | 287.0 | 261.4 | 227.7 | 184.2 | 150.8 |
| **Other farms** | 736.1 | 736.0 | 718.4 | 693.5 | 655.4 | 619.5 |
| **Total** | 1023.6 | 1023.0 | 979.8 | 921.2 | 839.6 | 770.3 |
| **% variation** | - | −0.1 | −4.3 | −10.0 | −18.0 | −24.7 |

Source: authors' own elaboration.

**Table A2.** Average gross margin per farm according to its technical orientation. Source: authors' own elaboration.

| Average GM per Farm (EUR) | s_land | s_nitrogen | s_em20 | s_em50 | s_em100 | s_em150 |
|---|---|---|---|---|---|---|
| **Dairy farm** | 88,816 | 88,625 | 80,719 | 70,327 | 57,415 | 48,025 |
| % variation | - | −0.21 | −9.12 | −20.82 | −35.36 | −45.93 |
| **Other farm** | 24,323 | 24,323 | 23,762 | 22,994 | 21,823 | 20,818 |
| % variation | - | 0.00 | −2.31 | −5.47 | −10.28 | −14.41 |

**Table A3.** Average gross margin per farm according to its technical orientation and class of age. Scenario "s_em150" is shown as the most representative carbon tax scenario.

| | s_land | | s_nitrogen | | s_em150 | |
|---|---|---|---|---|---|---|
| EUR/Farm | Dairy Farm | Other Farm | Dairy Farm | Other Farm | Dairy Farm | Other Farm |
| **≤40** | 34,570 | 34,580 | 34,442 | 34,581 | 17,196 | 28,071 |
| **41–64** | 98,968 | 30,260 | 98,924 | 30,256 | 56,466 | 25,433 |
| **≥65** | 93,782 | 17,279 | 93,316 | 17,279 | 47,138 | 14,982 |

Source: authors' own elaboration.

**Table A4.** Percentage variation in average gross margin compared to s_land.

| | s_nitrogen | | s_em150 | |
|---|---|---|---|---|
| % Variation | Dairy Farm | Other Farm | Dairy Farm | Other Farm |
| **≤40** | −0.37 | 0.00 | −50.26 | −18.82 |
| **41–64** | −0.04 | −0.01 | −42.94 | −15.95 |
| **≥65** | −0.50 | 0.00 | −49.74 | −13.29 |

Source: authors' own elaboration.

**Table A5.** Variation in number of dairy cows compared to s_land.

| Livestock Units | s_land | s_nitrogen | s_em20 | s_em50 | s_em100 | s_em150 |
|---|---|---|---|---|---|---|
| n. of dairy cows | 230,590 | 226,410 | 202,750 | 159,710 | 110,130 | 79,049 |
| % variation | - | −1.8 | −12.1 | −30.7 | −52.2 | −65.7 |

Source: authors' own elaboration.

**Table A6.** Nitrogen emission expressed in tonnes of nitrogen. Percentage variation is compared to s_land.

| Nitrogen Emission (t N) | s_land | s_nitrogen | s_em20 | s_em50 | s_em100 | s_em150 |
|---|---|---|---|---|---|---|
| t N | 22,959 | 22,535 | 20,196 | 15,925 | 11,031 | 7936 |
| % variation | | −1.8 | −12.0 | −30.6 | −52.0 | −65.4 |

Source: authors' own elaboration.

**Table A7.** Carbon emissions in thousand tonnes of $CO_2$eq.

| Carbon Emission (1000 t CO2eq) | s_land | s_nitrogen | s_em20 | s_em50 | s_em100 | s_em150 |
|---|---|---|---|---|---|---|
| Cereals | 317.2 | 316.5 | 302.3 | 293.4 | 282.5 | 272.4 |
| Forages | 227.2 | 227.4 | 244.1 | 245.5 | 247.1 | 248.6 |
| Maize | 160.2 | 158.9 | 136.9 | 109.3 | 77.0 | 55.4 |
| Proteic/Oilseeds | 61.9 | 63.5 | 64.6 | 65.0 | 64.8 | 64.4 |
| Meadows Pastures | 152.6 | 152.5 | 150.2 | 147.7 | 147.6 | 147.6 |
| Industrial crops | 131.1 | 126.1 | 86.1 | 75.3 | 66.2 | 60.9 |
| Milk | 1268.2 | 1245.2 | 1115.1 | 878.4 | 605.7 | 434.8 |
| Total | 2318.0 | 2290.0 | 2099.0 | 1815.0 | 1491.0 | 1284.0 |

Source: authors' own elaboration.

**Table A8.** Percentage variation in carbon emission compared to s_land.

| % Variation | s_nitrogen | s_em20 | s_em50 | s_em100 | s_em150 |
|---|---|---|---|---|---|
| Cereals | −0.2 | −4.7 | −7.5 | −11.0 | −14.1 |
| Forages | 0.1 | 7.4 | 8.0 | 8.7 | 9.4 |
| Maize | −0.8 | −14.5 | −31.8 | −51.9 | −65.4 |
| Proteic/Oilseeds | 2.6 | 4.5 | 5.0 | 4.7 | 4.1 |
| Meadows Pastures | −0.1 | −1.6 | −3.2 | −3.3 | −3.3 |
| Industrial crops | −3.8 | −34.3 | −42.6 | −49.5 | −53.5 |
| Milk | −1.8 | −12.1 | −30.7 | −52.2 | −65.7 |
| Total | −1.2 | −9.4 | −21.7 | −35.7 | −44.6 |

Source: authors' own elaboration.

**Table A9.** Water consumption in m$^3$.

| Water Consumption (Million m$^3$) | s_land | s_nitrogen | s_em20 | s_em50 | s_em100 | s_em150 |
|---|---|---|---|---|---|---|
| Cereals | 286.5 | 274.7 | 272.5 | 246.7 | 236.4 | 237.4 |
| Forages | 3335.1 | 3357.3 | 3642.9 | 3829.6 | 3983.7 | 4074.2 |
| Maize | 31.3 | 31.0 | 26.7 | 21.3 | 15.0 | 10.8 |
| Proteic/Oilseeds | 207.4 | 222.9 | 185.6 | 182.6 | 179.6 | 177.5 |
| Meadows Pastures | 610.5 | 609.8 | 600.8 | 590.7 | 590.5 | 590.3 |
| Industrial crops | 10.7 | 10.6 | 7.1 | 6.3 | 5.6 | 5.2 |
| Milk | 185.6 | 182.3 | 163.2 | 128.6 | 88.7 | 63.6 |
| Total | 4667.1 | 4688.6 | 4898.9 | 5005.9 | 5099.6 | 5159.1 |

Source: authors' own elaboration.

**Table A10.** Percentage variation in water consumption compared to s_land.

| % Variation | s_nitrogen | s_em20 | s_em50 | s_em100 | s_em150 |
|---|---|---|---|---|---|
| Cereals | −4.1 | −4.9 | −13.9 | −17.5 | −17.1 |
| Forages | 0.7 | 9.2 | 14.8 | 19.4 | 22.2 |
| Maize | −0.8 | −14.5 | −31.8 | −51.9 | −65.4 |
| Proteic/Oilseeds | 7.4 | −10.5 | −12.0 | −13.4 | −14.4 |
| Meadows Pastures | −0.1 | −1.6 | −3.2 | −3.3 | −3.3 |
| Industrial crops | −1.1 | −33.6 | −40.6 | −47.2 | −51.3 |
| Milk | −1.8 | −12.1 | −30.7 | −52.2 | −65.7 |
| Total | 0.5 | 5.0 | 7.3 | 9.3 | 10.5 |

Source: authors' own elaboration.

## Notes

[1]   The topic of this paper was presented as contributed paper at the 97th Annual Conference of the Agricultural Economics Society [1].

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
