# Peer review of "An Impact Assessment of GHG Taxation on Emilia-Romagna Dairy Farms through an Agent-Based Model Based on PMP"

_land, doi:10.3390/land12071409_

Round 1

Reviewer 1 Report

The manuscript evaluates the impact of carbon taxation on the dairy farms in an Italian province through agent based modeling. I have a few concerns about this study:

a. Other agent based models that study socioeconomic policy development need to be included. Example: Nisal A, Diwekar U, Hanumante N, Shastri Y, Cabezas H, Rico Ramirez V, Rodríguez-González PT. Evaluation of global techno-socio-economic policies for the FEW nexus with an optimal control based approach. Frontiers in Sustainability. 2022 Aug 31;3:948443. 

Also include other models that evaluate carbon taxation through mathematical modeling approaches.

b. Results: All the results are presented in a tabular format. These should be converted to figures to convey the impact of this study

c. The conclusions do not convey the significant impact of this study. This section needs to be modified to include key results, future work and takeaways from this study. 

d. Assumptions and limitations of this work need to be addressed

Author Response

The manuscript evaluates the impact of carbon taxation on the dairy farms in an Italian province through agent based modeling. I have a few concerns about this study:

We thank the reviewer for his/her comments that aim at further improving the quality of the paper. Hereafter, we reply point by point to all the comments. In the new version of the manuscript, the reviewer will find the punctual revisions marked in red.

  1. Other agent based models that study socioeconomic policy development need to be included. Example: Nisal A, Diwekar U, Hanumante N, Shastri Y, Cabezas H, Rico Ramirez V, Rodríguez-González PT. Evaluation of global techno-socio-economic policies for the FEW nexus with an optimal control based approach. Frontiers in Sustainability. 2022 Aug 31;3:948443. 

Integration line 120-134

ABMs are models composed by a set of decision makers (the agents) and an environment in which the agents interact with each other. They require rules to define the relationships between the agents and the relationships between agents and their economic and bio-physic environments, as well as rules defining the sequence of the actions occurring in the model [22]. Agent-based models are applied widely in many fields, such as consumer behaviour [23], travel forecasting [24] and diseases spreading control [25]. In agro-economics they have been heavily used in simulating land-use choices based on the agent’s utility derived from land [26,27] and as tool to explore the potential of landscapes to provide multiple ecosystem services [28]. The acting agent, with pre-defined behavioural rules set at the individual farmers’ level seems to be the appropriate starting point for explaining or predicting choices between different options [29]. According to Möhring, “the great achievement of agent-based models is their integration of the heterogeneity of individuals and transactions, accomplished by placing the optimisation process back on the unit where it actually occurs” ([29] page 10).

Integration line 137-155

In AGRISP, the agents are represented by individual farm holders, that exchange production factor, mainly land but also rights to pollute (nitrate spreading) [30]. The model optimizes simultaneously the production factors allocation of each agent, thus the decisions of each one of them impact the behaviour and the decisions of the others, simulating how their factors endowment evolves due to production factors exchange or policy impact. AGRISP captures farms heterogeneity in terms of farm structure and production strategies but also in terms of interactions between farms in the use of scarce resources and evaluate structural changes under the assumption of not-fully rational production choices, maximizing the utility function rather than the profit function [18,31]. Agents, and the environment in which they operate, are defined based on their characteristics. Agent’s attributes considered are the age of the farm holder and the presence of heirs. As far as the environment is concerned, altitude and agrarian regions are considered. Individual attributes trigger behavioural rules, more precisely:

  • farm holders older than 65 and with no successor receive a monthly retirement pension and do not rent additional land;
  • farmers located in NVZ limit manure spreading to 170 kg per hectare, whereas elsewhere limit is set to 340 kg/ha, and in both cases, farmers are pushed to rent out their land if unused.

AGRISP represents

Also include other models that evaluate carbon taxation through mathematical modeling approaches.

Integration line 167-168

such as AROPAj [38] are well suited to assess what production systems changes are needed to reduce GHG emissions [39] but

Integration line 175-178

Examples of well-established models, based on PMP and used for policy assessment, are IFM-CAP [43],which link emissions factors directly to the more granular defined production activities and FARMDYM which incorporates detailed emission accounting for different GHGs [44].

  1. Results: All the results are presented in a tabular format. These should be converted to figures to convey the impact of this study

Where meaningful, graphs have been added and related tables have been moved to an Appendix A.

  1. The conclusions do not convey the significant impact of this study. This section needs to be modified to include key results, future work and takeaways from this study. 

The conclusions section has been rewritten entirely to better convey the impact of this study.

  1. Assumptions and limitations of this work need to be addressed

Integration line 565 - 573

The model does not simulate the soil-plant interaction. Data on water consumption are taken from the FADN and C and N balances of various types of crops are not included in the calculation. Yields calculated in FADN are used to build the substitution matrix and the agricultural practices are not considered in detail.

The variable cost per crop, needed to build the Q matrix, are in this work estimated through a generalised least square approach. Even though these costs are available in the Italian FADN, we decided to opt for this methodological approach to compare the results with real data and define a method adoptable for other European regions where variable costs are not reported.

Integration line 577

from Pillar I

Integration line 578

Measures of Pillar II are not included in this work.

Integration line 579-592

Yet, the model only considers the dairy cow compartment, without considering the impact of other livestock, especially pig and poultry production, that still have a sub-stantial impact on GHG. The assumption that dairy cow livestock is closely linked to the available land, using fodder crops produced on farm, is taken to simplify the modelling of the rules imposed by the Parmigiano Reggiano Disciplinary [62]. The Disciplinary states that at least 50% of the dry matter of the fodder used must be produced on the farmland, and at least 75% of the dry matter of the fodder must be produced within the Parmigiano Reggiano cheese production area.

Integration line 603-610

Despite the decrease in number of LSU, the increase in land allocation of reused crops (forages and meadows and pastures) is due to the enforced agronomic constraint of equality between the total production of forage (including alfa-alfa, soja, protein crops, plus some industrial crops such as sugar beet and tomatoes) and total production of cereal (wheat, maize, barley and sorghum).

The nitrogen scenario does not seem to have a big impact on any of the indicators analysed, however the model proves to be suitable to simulate further restrictions that could be introduced in the regulation, given the persistence of the nitrogen issue.

We mean an increasing tax. Corrected with “increasing” in the document.

Reviewer 2 Report

This is a good study. The following comments aim to improve its quality.

1. It would be useful for the readers to explain the interaction of ABM and the PMP parts of the strategy.  Whilst the latter is sort of very well explained, the  former is not. Please describe the ABM in detail (e.g., what rules or assumptions you are considering behind).

2. Associated to the previous point, please add a diagram explaining the methodology. This will complement the previous point.

3. Motivate the choice of Emilia Romagna in your introduction and its importance on the Italian agriculture (e.g., can it be consider an example for the rest of the Italy?)

4. Your tables (table 5 onwards) can be improved. Instead of writing the variable's name as a heading consider using a proper name. The source of the tables should be below the table.  

In general, the text reads well but there are some typos (e.g., econometry instead of econometrics).

Author Response

This is a good study. The following comments aim to improve its quality.

We thank the reviewer for his/her comments that aim at further improving the quality of the paper. Hereafter, we reply point by point to all the comments. In the new version of the manuscript, the reviewer will find the punctual revisions marked in red.

(x) Minor editing of English language required: In general, the text reads well but there are some typos (e.g., econometry instead of econometrics).

Correction made on line 184

  1. It would be useful for the readers to explain the interaction of ABM and the PMP parts of the strategy.  Whilst the latter is sort of very well explained, the  former is not. Please describe the ABM in detail (e.g., what rules or assumptions you are considering behind).

Integration line 120-134

ABMs are models composed by a set of decision makers (the agents) and an environment in which the agents interact with each other. They require rules to define the relationships between the agents and the relationships between agents and their economic and bio-physic environments, as well as rules defining the sequence of the actions occurring in the model [22]. Agent-based models are applied widely in many fields, such as consumer behaviour [23], travel forecasting [24] and diseases spreading control [25]. In agro-economics they have been heavily used in simulating land-use choices based on the agent’s utility derived from land [26,27] and as tool to explore the potential of landscapes to provide multiple ecosystem services [28]. The acting agent, with pre-defined behavioural rules set at the individual farmers’ level seems to be the appropriate starting point for explaining or predicting choices between different options [29]. According to Möhring, “the great achievement of agent-based models is their integration of the heterogeneity of individuals and transactions, accomplished by placing the optimisation process back on the unit where it actually occurs” ([29] page 10).

Integration line 137-155

In AGRISP, the agents are represented by individual farm holders, that exchange production factor, mainly land but also rights to pollute (nitrate spreading) [30]. The model optimizes simultaneously the production factors allocation of each agent, thus the decisions of each one of them impact the behaviour and the decisions of the others, simulating how their factors endowment evolves due to production factors exchange or policy impact. AGRISP captures farms heterogeneity in terms of farm structure and production strategies but also in terms of interactions between farms in the use of scarce resources and evaluate structural changes under the assumption of not-fully rational production choices, maximizing the utility function rather than the profit function [18,31]. Agents, and the environment in which they operate, are defined based on their characteristics. Agent’s attributes considered are the age of the farm holder and the presence of heirs. As far as the environment is concerned, altitude and agrarian regions are considered. Individual attributes trigger behavioural rules, more precisely:

  • farm holders older than 65 and with no successor receive a monthly retirement pension and do not rent additional land;
  • farmers located in NVZ limit manure spreading to 170 kg per hectare, whereas elsewhere limit is set to 340 kg/ha, and in both cases, farmers are pushed to rent out their land if unused.

AGRISP represents

  1. Associated to the previous point, please add a diagram explaining the methodology. This will complement the previous point.

Integration line 216

Figure 1 is added to represent the general structure of the model.

  1. Motivate the choice of Emilia Romagna in your introduction and its importance on the Italian agriculture (e.g., can it be consider an example for the rest of the Italy?)

Integration line 45-47

The Emilia Romagna region is responsible for 10.4% of Italian livestock-related GHG emissions (2,059 thousand tonnes) [7], and its economy heavily relies on the Parmigiano Reggiano industry.

  1. Your tables (table 5 onwards) can be improved. Instead of writing the variable's name as a heading consider using a proper name. The source of the tables should be below the table.  

The name of the columns is the name of the scenarios, which are detailed in the scenario description section. The variables considered are explained in the table title. Source of the tables has been moved below each table.

We mean an increasing tax. Corrected with “increasing” in the document.

Reviewer 3 Report

This article aims at estimating the impacts of an environmental tax from the structural, environmental, productive and economic point of view. The issue is clearly politically relevant and assessment of the likely impacts of policy measures is in acute need.  The methodology, integrating an Agent Based Model (ABM) and Positive Mathematical Programming (PMP) is sophisticated and innovative. Overall, this is a very good paper and I only have one main clarification to ask, and few secondary remarks or suggestions.

1.      The constraint that “the livestock is closely linked to the available land, through the use of fodder crops produced on farm” seems to me a rather strong assumptions, since in practice farmers con buy and exchange fodder, easier than exchanging land. I am not particularly familiar with PMP models, so I do not know if the constraint (eqn. 4) is needed for the model to work. Regardless, my feeling is that this should be discussed, and the possible limitations of the model due to this assumption should be acknowledged.

Secondary remarks.

2.      One scenario concerns the introduction of the Nitrate Directive. But the Directive has been implemented in Emilia Romagna in 2017, while the sample refers to 2020. So, in principle farms in 2020 should have already complied with the Directive. Regardless, the explanation of the cost at lines 370-376 is unclear “Dairy farmers [exceeding the limit] can acquire rights to pollute and spread the excess manure from non-livestock farms that need nitrogen fertilizers, paying a cost of 150 €/ha. The distribution cost of excess manure is set to 69€/ton nitrogen, based on the average price (80 €/hour) and capacity (4.5 tonnes of manure) of a manure tank, and the nitrogen content of dairy cow manure (0.42%)”: the distribution cost (69€/ton nitrogen) adds to the cost of 150€/ha? And how this relate to the cost of a manure tank?

3.       Table 4: it is surprising that water requirements for cereals are higher than for maize: please check

4.      In Table 6, the number of farms can have no decimal

5.      In Table 9, “other farms” below 40 ha increase their income following the introduction of the Nitrate Directive: is this because they sell rights to spread the manure? In general, some speculation on the reasons of the results would be useful

6.      Line 478 “The increase in grazing meadows is justified by the farmers' strategy of moving towards those crops that retain less CO2 in a strategy of progressive extensification.” Maybe it should be “that retain more CO2” or that “emit less CO2

7.      You use several times the term “progressive carbon tax”. Do you really hypothesize a progressive tax in the sense that tax rates grow with the amount of emissions? If so, please explain the rates, otherwise the term could be misleading.

Author Response

This article aims at estimating the impacts of an environmental tax from the structural, environmental, productive and economic point of view. The issue is clearly politically relevant and assessment of the likely impacts of policy measures is in acute need.  The methodology, integrating an Agent Based Model (ABM) and Positive Mathematical Programming (PMP) is sophisticated and innovative. Overall, this is a very good paper and I only have one main clarification to ask, and few secondary remarks or suggestions.

We thank the reviewer for his/her comments that aim at further improving the quality of the paper. Hereafter, we reply point by point to all the comments. In the new version of the manuscript, the reviewer will find the punctual revisions marked in red.

  1. The constraint that “the livestock is closely linked to the available land, through the use of fodder crops produced on farm” seems to me a rather strong assumptions, since in practice farmers can buy and exchange fodder, easier than exchanging land. I am not particularly familiar with PMP models, so I do not know if the constraint (eqn. 4) is needed for the model to work. Regardless, my feeling is that this should be discussed, and the possible limitations of the model due to this assumption should be acknowledged.

Integration line 581-599

The assumption that dairy cow livestock is closely linked to the available land, using fodder crops produced on farm, is consistent with the usual farm practice, according to which all fodder production obtained on the farmland is addressed to feed livestock and with the rules imposed by the Parmigiano Reggiano PDO Code of Specification [62]. The Code of Specification states that at least 50% of the dry matter of the fodder used must be produced on the farmland, and at least 75% of the dry matter of the fodder must be produced within the Parmigiano Reggiano cheese production area. The strict linkage between the feed requirement and farmland allows to impose an implicit structural limit on the animal production capacity at the farm level and to merge land cultivation and livestock in one single farmer’s optimisation strategy. In other words, for dairy farms, the hypothesis is to maximise the value of the milk production intended as transformation from fodder to milk. In this context, the dairy farm is pushed to first employ all fodder produced in-farm until the economic equilibrium condition is fulfilled: marginal cost of milk production, represented by the fodder (and feed) marginal cost is equal to marginal revenue (milk price). It is worth noting that concentrates and off-farm hay procurement are not missing in the model, but they are included as a component of the milk production cost. Therefore, one of the main characteristics of the livestock PMP approach in AGRISP is the absence of the fodder consumption function based on technical coefficients, which is replaced by the cost function.

Secondary remarks.

  1. One scenario concerns the introduction of the Nitrate Directive. But the Directive has been implemented in Emilia Romagna in 2017, while the sample refers to 2020. So, in principle farms in 2020 should have already complied with the Directive.

Integration line 421-424

The reason why we consider this scenario is to estimate a priori, based on the number of LSU and the land endowment of the dairy farms, whether the Nitrogen Directive in Emilia Romagna can be respected or if dairy farmers must search for manure disposal solutions outside of their farm boundaries.

… Regardless, the explanation of the cost at lines 370-376 is unclear “Dairy farmers [exceeding the limit] can acquire rights to pollute and spread the excess manure from non-livestock farms that need nitrogen fertilizers, paying a cost of 150 €/ha. The distribution cost of excess manure is set to 69€/ton nitrogen, based on the average price (80 €/hour) and capacity (4.5 tonnes of manure) of a manure tank, and the nitrogen content of dairy cow manure (0.42%)”: the distribution cost (69€/ton nitrogen) adds to the cost of 150€/ha? And how this relate to the cost of a manure tank?

See corrections between line 402 and 421

We try to better explain the calculation done.

Integration line 608-610

The nitrogen scenario does not seem to have a big impact on any of the indicators analysed, however the model proves to be suitable to simulate further restrictions that could be introduced in the regulation, given the persistence of the nitrogen issue.

  1. Table 4: it is surprising that water requirements for cereals are higher than for maize: please check

Thanks for pointing this out. We have further disaggregated table 3 and 4, as we initially included also rice among cereals, which have a very high WFP. In addition, please note that the WFP calculation also considers the fact that cereals have a longer period of permanence in the soil, hence bigger water need on a per m3/year basis.

  1. In Table 6, the number of farms can have no decimal

corrected.

  1. In Table 9, “other farms” below 40 ha increase their income following the introduction of the Nitrate Directive: is this because they sell rights to spread the manure? In general, some speculation on the reasons of the results would be useful

We reviewed the calculation, and we discover a small mistake. Thanks to point it out. See updated table in the Appendix (Table 3). Now the difference is too small (1€) to speculate on it.

  1. Line 478 “The increase in grazing meadows is justified by the farmers' strategy of moving towards those crops that retain less CO2 in a strategy of progressive extensification.” Maybe it should be “that retain more CO2” or that “emit less CO2

Corrected in line 514, indeed it was a mistake.

  1. You use several times the term “progressive carbon tax”. Do you really hypothesize a progressive tax in the sense that tax rates grow with the amount of emissions? If so, please explain the rates, otherwise the term could be misleading.

We mean an increasing tax. Corrected with “increasing” in the document.
